# Natural Killer Cells Are Key Host Immune Effector Cells Affecting Survival in Autologous Peripheral Blood Hematopoietic Stem Cell Transplantation

**DOI:** 10.3390/cells11213469

**Published:** 2022-11-02

**Authors:** Luis F. Porrata

**Affiliations:** Division of Hematology, Department of Internal Medicine, Mayo Clinic, Rochester, MN 55902, USA; porrata.luis@mayo.edu; Tel.: +1-507-284-5096

**Keywords:** natural killer cells, autologous peripheral blood hematopoietic stem cell transplantation, survival

## Abstract

The infusion of autograft immune effector cells directly impacts the clinical outcomes of patients treated with autologous peripheral blood hematopoietic stem cell transplantation, suggesting the possibility of an autologous graft-versus tumor cells. Furthermore, the early recovery of immune effector cells also affects survival post-autologous peripheral blood hematopoietic stem cell transplantation. Natural killer cells are among the immune effector cells reported to be collected, infused, and recovered early post-autologous peripheral blood hematopoietic stem cell transplantation. In this review, I attempt to give an update on the role of natural killer cells regarding improving survival outcomes on patients treated with autologous peripheral blood hematopoietic stem cell transplantation.

## 1. Introduction

The concept of an autologous graft-versus-tumor effect is changing the landscape of recognizing the autologous peripheral blood hematopoietic stem cell transplantation (APBHSCT) as an immunotherapeutic intervention rather than a way to infuse high-doses of chemotherapy with the hope that the high doses of chemotherapy will eradicate cancer cells [1]. Not only the infusion of autograft immune effector cells but also the recovery of those infused immune effector cells have been associated with survival outcomes post-APBHSCT [1]. Among the immune effector cells collected and infused in conjunction with stem cells in APBHSCT are natural killer (NK) cells. The infusion of autograft NK cells has been specifically associated with superior overall survival (OS) and progression-free survival (PFS) for patients treated with APBHSCT. Furthermore, the sustained recovery of NK cells improves clinical outcomes post-APBHSCT. Thus, in this article, I review our current knowledge of the role that NK cells play in improving survival outcomes in patients undergoing APBHSCT.

## 2. Natural Killer Cell Biology

NK cells comprise 5% to 8% of the human peripheral blood lymphocytes, approaching two billion NK cells circulating in human adults [2]. NK cells are an integrated component of the innate immune system and belong to the family of the innate lymphoid cells. In humans, NK cells are divided into two main subsets: CD56^bright^ CD16^dim^ and CD56^dim^ CD16^bright^ [3]. CD56^dim^ CD16^bright^ natural killer cells contain higher number of cytolytic granules, such as perforin and granzyme A and B [3]. In contrast, CD56^bright^ CD16^dim^ NK cells promote immune regulation by cytokine secretion including interferon-gamma, tumor necrosis factor-alpha and beta, and granulocytic macrophage-colony stimulating factor [3]. CD56^bright^ CD16^dim^ NK cells have a weaker cytotoxicity ability compared with CD56^dim^ CD16^bright^ NK cells [3].

### 2.1. Natural Killer Cell Receptors and Their Effector Function

The ability of NK cells to recognize and eliminate tumor cells is controlled by different type of receptors expressed on the NK cell surface. The first group of these receptors falls under the description of inhibitory receptors such as killer cell immunoglobulin-like receptors (KIRs). KIRs are inactivate NK cells upon binding to classical major histocompatibility complex class I molecules HLA-A, -B, and -C [4]. A second group of inhibitory receptors comprise the C-type lectin-like heterodimeric receptor natural killer group 2 member C (NKG2A and its isoform NKG2B) that recognize the non-classical major histocompatability complex class I molecule HLA-E [5].

The activating NK cell receptors include natural killer group 2 member C (NKG2A), NKG2D, signaling lymphocyte activation molecule (SLAM) family molecule 2B4 (CD244), the co-activating receptor/adhesion molecule DNAX accessory molecules-1 (DNAM-1, CD226), and the natural cytotoxicity receptors (NCRs) (NKp30, NKp44, NKp46, and NKp80). NKG2D receptor on NK cells can bind on tumor cells expressing major histocompatibility complex class I A and B molecules, leading to tumor death [6]. The platelet-derived growth factor (PDGF) secreted by tumor cells can be recognized by NKp44 triggering NK cell secretion of interferon-gamma and tumor necrosis factor-alpha to hamper tumor growth [7]. NK cells might limit tumor metastasis by activating NKp46 in the intratumoral NK cells secreting interferon-gamma, and interferon-gamma increases the expression of extracellular matrix protein fibronectin 1, affecting the primary tumor architecture [8].

Adaptive NK cell is a subset of NK cells with adaptive immune features including memory-like properties, such as long-term persistence [9], robust preferential expansion in response to viral infection (i.e., cytomegalovirus) [10], and antibody-dependent effector function [11]. Adaptive NK cells express the maturation marker CD57 and the activating receptor NKG2C [12]. Furthermore, adaptive NK cells are resistant to tumor microenvironment immunosuppressive agents such as regulatory T-cells [13] and myeloid-derived suppressor cells (MDSC) [14], which are involved in the pathogenesis of cancer [14].

### 2.2. Natural Killer Cells’ Mechanisms of Action of Eradicating Cancer Cells

NK cells play a significant role in immunosurveillance. In an 11-year follow-up study, it was found the patients with low NK cell cytotoxicity in an analysis of peripheral-blood mononuclear cells correlated with an increase in cancer development. Using low cytotoxic activity as a reference, the relative risk was 0.72 (95% CI 0.45–1.16) for men with high cytotoxic activity and 0.62 (95% CI 0.38–1.03) for men with medium cytotoxic activity. For women with high cytotoxic activity, the relative risk was 0.52 (95%CI 0.28–0.95), and for medium cytotoxic activity, it was 0.56 (95%CL 0.31–1.01). For both genders with high and medium cytotoxic activity, the relative risk was 0.63 (95%CI 0.43–0.92) for men and 0.56 (95%CI 0.40–0.87) for women [15]. The mechanisms of action of NK cells for immunosurveillance and killing of tumor cells can be divided into direct killing and indirect killing.

By direct cancer killing, transformed cells that present with lower expression or mismatch expression of major histocompatibility complex class I molecules trigger activation of NK cells. Upon activation, NK cells secrete perforin and granzyme to eradicate tumor cells [16]. NK cells also express tumor necrosis factor family member such as Fas/Apo-1/CD95 ligand (FasL) or tumor necrosis factor related apoptosis by direct interaction with their respective receptors Fas and TRAIL receptor (TRAILR) on cancer cells [17]. NK cells can kill cancer cells by CD16A(FcgRIIIA)-mediated antibody-dependent cellular toxicity (ADCC), as NK cells can act a bridge between the anti-tumor antibodies IgG1 and IgG3, whereby Fab specifically recognizes the tumor while the Fc segment bind to the natural killer cell [18]. The CD16A can also enhance the interleukin-2 (IL-2) and interleukin-15 (IL-15)-driven NK cell proliferation and cytotoxicity against tumor cells [19].

By indirect killing, NK cells can activate immune effector cells from the innate and adaptive immune systems as well as inhibiting tumor angiogenesis [20,21]. For example, NK cells can prevent tumor-infiltrating neutrophils from helping with tumor angiogenesis by secreting vascular endothelial growth factor (VEGF) in the tumor microenvironment [22]. NK cells can modulate dendritic cell antigen presentation in the adaptive immune system by presenting cancer antigens to the dendritic cells from the cancer cells killed by NK cells [23]. Stimulated NK cells secreting interferon-gamma can trigger the transformation of CD8+T-cells into cytotoxic T-cell lymphocytes and the transformation of CD4+Tcells into T-cell helper 1 cells [24]. T-cell helper 1 cells can further promote cytotoxic T-cell lymphocytes.

## 3. B-Cell, T-Cell, and Natural Killer Cell Reconstitution Post-Autologous Peripheral Blood Hematopoietic Stem Cell Transplantation

In comparison to allogeneic stem cell transplantation, which requires immunosuppressive therapy to prevent host versus graft rejection and to control graft-versus-host disease, APBHSCT provides a cleaner picture of immune reconstitution after stem cells are infused as no immunosuppressive therapy is needed post-APBHSCT that could alter the quantitative and qualitative recovery of immune effector cells after stem cell/bone marrow transplantation. Even though neutrophil and platelet recovery is often considered the end point of hematologic engraftment after APBHSCT, the immunologic reconstitution occurs gradually, and the normal humoral and cellular immune recovery could take up to one year after APBHSCT [25].

### 3.1. B-Cell Reconstitution

B-cell recovery after APBHSCT recapitulates normal B-cell ontogeny. The relative and absolute number of circulating B-cells expressing CD19 and CD20 consistently decreased within the first 3 months and can remain low as long as 18 months after APBHSCT [26]. The deficient functional recovery of B-cells after APBHSCT is attributed to the decrease in T-cell help and intrinsic B-cell defects [27,28]. Normal serum immunoglobulin B-cell production could take 6 months for IgM, 12 to 18 months for IgG, and years for IgA [26].

### 3.2. T-Cell Reconstitution

Quantitative recovery after APBHSCT has shown low levels of relative and absolute numbers of CD3 between 3 and 5 months [28], CD4 lasting for a year or longer [29], and CD8 varying between 3 and 18 months [28]. Some reports have shown low levels of CD3 and CD4, leading to an inverted CD4/CD8 ratio for up to 10 years after autologous stem cell transplantation.

The normal functionality of T-cell recovery is also delayed post-autologous stem cell transplantation. T-cell proliferation, T-cell cytokine production, T-cell response to exogenous IL-2, and T-cell cytotoxicity have been reported to be decreased as early as 2 months up to 5 years post-autologous stem cell transplantation [29].

### 3.3. Natural Killer Cell Reconstitution

The quantitative recovery of NK cells has been shown to be faster between 3 and 5 weeks in the autologous stem cell transplantation compared with the allogeneic stem cell transplantation. Furthermore, NK cell recovery by day 30 post-autologous stem cell transplantation was faster after APBHSCT in comparison to patients that underwent bone marrow harvest for their autologous stem cell transplantation [28]. Currently, peripheral stem cell collection is the preferred method compared with bone marrow harvest to collect CD34 stem cells prior to autologous stem cell transplantation. The earliest that natural killer cell recovery has been reported is around day 14 after autologous stem cell transplantation [25]. Talmadge et al. reported that, by day 75 after APBHSCT, the NK cell activity was the same as normal controls (20–25% cytotoxicity by chromium-51 cytotoxic assay) [29]. Our group reported that NK cell activity can be detected as early as day 14 after autologous stem cell transplantation. Day 14 after autologous stem cell transplantation, NK cells were collected and incubated with IL-2 and interferon-gamma. Increased NK cell activity was observed by increased lysis of K562 cells from NK cells incubated with IL-2 and interferon-gamma compared with placebo (effector to target ratio, 50:1, *p* < 0.001) [30]. The early NK cell function recovery after autologous stem cell transplantation has been recently confirmed, showing the early recovery of NK cells were able to degranulate (CD107a expression) and produce interferon-gamma and macrophage inflammatory protein-1β upon tumor interaction [31]. Thus, the early quantitative and qualitative NK cells recovery argues in favor of viewing NK cells as useful immunomodulators as early as day 14 post-APBHSCT to eradicate or control minimal residual disease.

## 4. Infusion of Autograft-Natural Killer Cells and Survival

Our group first reported that the infusion of autograft-absolute lymphocyte count (A-ALC) directly affected clinical outcomes post-APBHSCT [32,33]. In a randomized, double-blind phase clinical trial to confirm the prognostic ability of A-ACL, a subset analysis of the immune effector cells collected and infused identified for the first time that the infusion of low number of autograft NK cells negatively affected OS (hazard ratio (HR) = 2.08, 95% confidence interval (CI), 1.12–3.87, *p* < 0.02) and PFS (HR = 2.17, 95% CI, 0.97–4.84, *p* = 0.06) post-APBHSCT [34]. Our group reported an up-date of the survival benefit on the infusion of autograft natural killer cells in the patients accrued to our phase III clinical trial. With a median follow-up of 5.2 years, patients infused with autograft NK cells ≥ 0.09 × 10^9^ cells/kg experienced a 5-year OS rates of 87% and 5-year PFS rates of 71% compared with patients infused with autograft NK cells < 0.09 × 10^9^ cells/kg with 5 years rates of 55% and 32%, respectively [35]. With a final up-date of a 10.6 year median follow-up, the 13 years OS rates for the group infused with autograft NK cells ≥ 0.09 × 10^9^ cells/kg were 46% versus 36% for the group infused with autograft NK cells < 0.09 × 10^9^ cells/kg, *p* < 0.02. The 13 years PFS rates were 45% for the autograft NK cells ≥ 0.09 × 10^9^ cells/kg group compared with 21% for the autograft NK cells group, *p* < 0.0004 [36]. To our knowledge, our latest report is the longest published survival data showing superior OS and PFS by collecting and infusing autograft natural killer cells in APBHSCT [36].

In a multicenter prospective study entitled the graft and outcome in autologous stem cell transplantation (GOA), Turunen et al. reported that the infusion of high number or autograft NK cells were a negative prognostic factor for PFS in non-Hodgkin’s lymphoma patients undergoing autologous stem cell transplantation. However, looking at the Kaplan–Meier survival curves, the differences in survival between the group that had lower autograft NK cells compared with the group that had higher number of autograft NK cells showed that the 5-year OS rates were approximately 80% for the patients who received autograft NK cells < 0.5 × 10^6^ cells/kg, versus 70% for the patients infused with autograft NK cells ≥ 0.5 × 10^6^ cells/kg (*p* = 0.092), and the median follow-up for both groups has not been reached. The 5-year PFS rates were approximately 75% for the patients received autograft NK cells < 0.5 × 10^6^ cells/kg versus 55% for the patients infused with autograft NK cells ≥ 0.5 × 10^6^ cells/kg (*p* = 0.048), and the median follow-up for both groups have not been reached [37]. In contrast, the same group reported superior overall survival in non-high risk cytogenetics multiple myeloma patients that received autograft NK cells ≥ 2.5 × 10^6^ cells/kg (the 5-year OS rates approximately 95%) compared with patients that received autograft NK cells < 2.5 × 10^6^ cells/kg (the 5-year OS rates approximately 65%) (*p* < 0.006) [38].

### Autograft Natural Killer Cell Receptors and Survival

To further understand how the infusion of the autograft NK cells impacts clinical outcomes, our group investigated the autograft NK cell receptors and their association with survival post-APBHSCT. Looking into the killer cell immunoglobulin-like inhibitory receptors, our group reported that the infusion of higher numbers of KIR2DL2 was associated with worse prognosis post-APBHSCT. Patient infused with autograft NK KIR2DL2 cells ≥ 0.066 × 10^9^ cells/kg experienced 5-year OS rates of 44% with a median follow-up of 48.21 months compared with patients infused with autograft KIR2DL2 cells < 0.066 × 10^9^ cells/kg experiencing 5-year OS rate of 88% with a median follow-up that has not been reached (*p* < 0.0001) [39]. Looking into the natural cytotoxicity receptors, our group reported that the infusion of higher numbers of NKp30 was associated with better prognosis post-APBHSCT. Patient infused with autograft NKp30 cells ≥ 0.09 × 10^9^ cells/kg experienced 5-year OS rates of 95% with a median follow-up that has not been reached compared with patients infused with autograft NK NKp30 cells < 0.09 × 10^9^ cells/kg experiencing 5-year OS rates of 34% with a median follow-up of 39.80 months (*p* < 0.0001) [39]. Furthermore, autograft NKp30 cells also affected PFS. Patient infused with autograft NKp30 cells ≥ 0.09 × 10^9^ cells/kg experienced 5-year PFS rates of 77% with a median follow-up that has not been reached compared with patients infused with autograft NKp30 cells < 0.09 × 10^9^ cells/kg experiencing 5-year PFS rates of 13% with a median follow-up of 12.67 months (*p* < 0.0001) [39].

## 5. Survival Based on Absolute Natural Killer Cell Count Recovery Post-Autologous Peripheral Blood Hematopoietic Stem Cell Transplantation

The recovery of the absolute lymphocyte count at day 15 (ALC-15) ≥ 500 cells/µL after autologous stem cell transplant has been reported to be an independent predictor for OS and PFS [40].

Since NK cells are the first immune effector cells that recovered both quantitatively and qualitatively as early as day 14 post-APBHSCT, we evaluated prospectively if the natural killer cells were the key lymphocytes conveying the survival advantage in patients with an ALC-15 ≥ 500 cells/µL. By univariate analysis, NK cells were the only lymphocyte subset in the ALC-15 to be a predictor for survival post-APBHSCT. Patients that recovered NK cells ≥ 80 cells/µL by day 15 post-APBHSCT experienced superior OS and PFS compared with those who did not (median OS was not reached versus 5 months, 3-year OS rates of 76% versus 36%, *p* < 0.0001; median PFS was not reached versus 3 months, 3-year PFS rates of 57% versus 9%, *p* < 0.0001, respectively) [41]. Rueff et al. reported in multiple myeloma patients that at 1 month post-autologous stem cell transplantation compared with patients with low NK cells (<100 cells/µL), high natural killer cell counts were associated with better PFS: for NK cells between 100 and 200 cell/µL, the hazard ratio was 0.33, 95% CI, 0.16–0.80, *p* < 0.004, and for NK cells > 200 cells/µL, the hazard ratio was 0.27, 95% CI, 0.15–0.58, *p* < 0.001 [42].

Another biomarker in multiple myeloma associated with survival is the achievement of negative minimal residual disease. Paiva et al. reported prolonged OS and PFS in multiple myeloma patients who achieved a negative minimal residual disease by day 100 compared with those who did not post-autologous stem cell transplantation [43]. Thus, several studies have investigated the association between immune effector cells recovery and minimal residual disease in multiple myeloma post-APBHSCT. Keruakos et al. reported an absolute mean number of NK cells higher (median 131.38 cells/µL) in patients that achieved a negative minimal residual disease compared with those who did not achieve minimal residual disease due to lower absolute mean number of natural killer cells (median 43.36 cells/µL) two–three months post-APBHSCT. Furthermore, the odd ratio was 7.5 higher (95% CI: 5.21–9.79) to achieve a negative minimal residual disease with normal NK cell numbers (≥76 cells/µL) compared with a lower number of NK cells (<76 cells/µL) [44].

Similarly, Bhutani et al. reported in 36 multiple myeloma with minimal residual disease analysis post-APBHSCT a lower proportion of circulating natural killer cells in positive minimal residual disease patients compared with negative minimal residual disease patients (6.1% ± 0.6% versus 10.8% ± 3.6%, respectively). In addition, 12 of the 30 positive minimal residual disease patients that relapsed had lower NK cells counts (3.9 ± 2.0 versus 7.5 ± 3.3, *p* < 0.0019) compared with patients who did not progress during the study follow-up time [45].

In non-Hodgkin’s lymphoma, sustaining an absolute number of NK cells ≥250 cells/µL at 3, 6, 9, and 12 months post-APBHSCT was an independent predictor for OS (HR = 0.013, 95%, CI 0.001–0.063, *p* < 0.0001) and PFS (HR = 0.014, 95%CI, 0.001–0.067, *p* < 0.0001) [46]. These studies strongly suggest that the quantitative recovery of NK cells post-APBHSCT affect clinical outcomes.

## 6. Survival Based on Natural Killer Cell Receptor Recovery Post-Autologous Peripheral Blood Hematopoietic Stem Cell Transplantation

Jacobs et al. reported that, during leukocyte engraftment after APBHSCT, CD56^bright^ NK cells were the major NK cell subset identified. These CD56^bright^ NK cells showed high surface expression of CD57 and killer Ig-like receptors with up-regulation of KIR2DL2/3/S2 and KIR3DL1, whereas KIR2DL1/S1 remained constant [47]. This was a descriptive and functionality study of NK cell recovery post-APBHSCT. In another study assessing multiple myeloma patients that remained in complete remission more than 6 years after APBHSCT, Arteche-Lopez et al. reported fewer activating receptors such as NKp46 together with increased expression of inhibitory NKG2A and KIR2DL1 receptors [48]. A possible explanation was that, early post-APBHSCT, continuous cytotoxic activity against myeloma cells is driven by the activating signals of natural killer cells to achieve complete remission, leading to downregulation of the these activating signals and increased expression of inhibitory receptors (NKG2A and KIR2DL1), confirming the characteristic phenotype terminally differentiated natural killer cells [48].

Cytomegalovirus seropositive multiple myeloma with high numbers of adaptive (NKG2C) NK cells (>1.58µL) at day 28 after APBHSCT was associated with decreased risk of relapse (at 2 years, relapse rates of 22% with high adaptive NK cell group compared with 75% in the low adaptive natural killer cell group (*p* < 0.0062)) [49].

## 7. Mismatch between KIR Expression and HLA Class I Ligands in Autologous Stem Cell Transplantation

In the allogeneic stem cell transplantation, the graft-versus-tumor effect exerted by NK cells relies on the mismatch between KIR expression by donor NK cells and the HLA I ligands expressed on the recipient cancer cells [50]. Inhibitory KIR–HLA–receptor–ligand mismatch can also occur in autologous stem cell transplantation. Leung et al. reported that, in a cohort of lymphoma and solid tumor patients, the cumulative incidence of disease progression at 1000 days after autologous stem cell transplantation was 0% with 2 mismatch pairs, 50% with 1 mismatch pair, and 83% with zero mismatch pairs (*p* < 0.01) at 1000 days post-autologous stem cell transplantation [51]. In patients with acute myeloid leukemia treated with autologous stem cell transplantation, Marra et al. reported that patients with a compound KIR3DL1+, HLA-Bw4-80Thr+, and HLA-Bw4-80Ile- genotype, predictive of low-affinity interactions, had a low incidence of relapse, compared with patients with a KIR3Dl1+ and HLA-Bw4–80ILe+ genotype, predictive of high-affinity interactions (HR = 0.22, 95% CI, 0.06–0.78, *p* < 0.02) [52]. This low incidence of relapse was influenced by HLA-Bw4 copy number, such that relapse progressively increased with one copy of HLA-Bw4-80Ile (HR = 1.6, 95%CI, 0.84–3.1, *p* = 0.15) to two to three copies (HR = 3.0, 95% CI, 1.4–6.5, *p* < 0.005) and progressively decreased with one to two copies of the HLA-Bw4-80Thr. Patients with KIR3DL1+ and HLA-Bw4-80ILe+, predicted of a low-affinity KIR2DL2/3 and HLA-C1/C1 genotype, was associated with lower relapse than a predicted high-affinity KIR2DL1+ and HLA-C2/C2 genotype (HR = 0.25, 95%CI, 0.09–0.73, *p* < 0.01). Additionally, a KIR3DL1+ and HLA-Bw4-80Ile+ genotype rescued KIR2DL1+ and HLA-C2/C2 patients from high relapse (*p* < 0.007). The authors argue that these findings support the role of NK cell graft-versus-leukemia activity modulated by NK cell receptor–ligand affinities in autologous stem cell transplantation for patients with acute myeloid leukemia. Multiple myeloma patients with KIR3DS1+, KIR3DL1+, HLA-Bwa had shorter progression-free survival than those who were KIR3DS1- (12.2 versus 24 months, respectively, *p* < 0.002) post-autologous stem cell transplantation [53]. High-risk neuroblastoma patients treated with autologous stem cell transplantation showed 46% lower risk of death and 34% lower risk of progression at 3 years for patients lacking an HLA ligand compared with patients possessing all of the ligands for their inhibitory KIR [54]. In NHL, an HLA-Cw8 genotype was associated with worse overall survival [55].

## 8. Natural Killer Cell Immunometabolism and Autologous Peripheral Blood Hematopoietic Stem Cell Transplantation

Excellent reviews of NK cell immunometabolism have been published [56]; thus, in this section, I provide a summary of the important findings of NK cell immunometabolism and how they apply in the APBHSCT setting. Glucose is transported to NK cells via glucose transporter 1 (GluT1), which is upregulated by NK cells after activation. Glucose is then metabolized by glycolysis to produce pyruvate. Pyruvate undergoes metabolism to lactate, which is secreted by the cell, or pyruvate enters the mitochondria where most of it is metabolized by the citrate malate shuttle (CMS) and the TCA cycle (Kreb’s cycle), thus fueling oxidative phosphorylation (Oxphos) and leading to efficient production of ATP. Warburg discovered that cancer cells, even in the presence of oxygen, preferentially metabolized glucose to lactate in a process (due to the redundant presence of oxygen) known as aerobic glycolysis or Warburg metabolism. In normal cells, aerobic glycolysis not only is used as a response to a lack of oxygen but also is a bone fide pathway on its own [56]. Other sources of energy include glutamine and fatty acids. Glutamine feeds into the TCA cycle after converting into the TCA cycle intermediate α-Ketoglutarate. Fatty acids feed into the TCA cycle after fatty acid oxidation (FAO) generates the TCA cycle intermediate acetyl-CoA [57].

Resting NK cells utilize Oxphos to meet their homeostasis needs. When NK cells become activated using IL-2, IL-12, IL-15, and IL-18, NK cells strongly upregulate both the glycolysis and Oxphos pathways [57]. CD56^bright^ NK cells are more prone to increase glycolytic metabolism [58]. Lipid accumulation alters the metabolism responses of NK cells, and it can lead to the inhibition of NK cell effector functions. The inhibition of carnitine palmitoyl-transferase 1 (CPT1), a critical transporter for the oxidation of long-chain fatty acids within the mitochondria using etomoxir, could restore glycolytic activity and cytotoxicity of NK cells [58].

Richter et al. evaluated the metabolic activity of engrafted NK cells post-APBHSCT [59]. In this study, engrafted NK cells showed high expression of hexokinase 2 (HK2), the rate-limiting enzyme of glycolysis. This was accompanied by an abundance of Glut1 and increased glucose uptake. Furthermore, CPT1, the rate-limiting enzyme of the FAO pathway, was significantly increased by NK cells and the increased CPT1 expression during engraftment of NK cells was accompanied by an increased uptake of fluorescent fatty acids analogues. In the study, NK cells expressing higher levels of CPT1 (>2830 modified fluorescence intensity (mdFI)) were associated with worse progression survival. This might be explained as CPT1 has been associated with impaired NK cells cytotoxicity. Additionally, a higher expression of HK2 in engrafted NK cells (>1414 MdFI) was associated with worse progression survival. However, the underlying mechanism of engrafted NK cells expressing elevated levels of HK2 and worse survival post-APBHSCT is still unknown.

Table 1 summaries the clinical outcomes based on NK cells post-APBHCST.

## 9. Engineering an Autograft Natural Killer Cell and Enhancing Natural Killer Cell Recovery/Engraftment versus Tumor Effect in Autologous Peripheral Blood Hematopoietic Stem Cell Transplantation

Since the quantitative collection and infusion of autograft NK and recovery of NK cells post-APBHSCT directly correlates with survival, methods to increase the peripheral blood mobilization of NK cells numbers for collections and improves NK cell recovery post-APBHSCT are warranted.

### 9.1. Methods to Enhance the Collection of Autograft NK Cells

The number of apheresis collections: Our group identified lymphoma patients who underwent four or more apheresis collections of autograft NK compared with lymphoma patients who had less than four apheresis collections (≥4 collections: median autograft NK cells = 0.07 × 10^9^ cells/kg (range: 0.002–0.21 × 10^9^ cells/kg, versus less than 4 collections: median autograft NK cells = 0.05 × 10^9^ cells/kg (range: 0.007–0.17 × 10^9^ cells/kg), *p* < 0.004) [60]. However, major drawbacks of this method are cost and patient convenience.

Maximizing apheresis machines’ lymphocyte/NK cell collection: Our apheresis group published that changing the mononuclear cell (MNC) offset of 1.5 and that of red blood cells (RBCs) of 5.0 to a new setting on the Fenwal Amicus machine of 1.5 for mononuclear cells and of 6.0 for red blood cells (RBCs) resulted in higher numbers of A-ALC collected [61]. Jalowiec et al. recently reported that performing high-flow lymphocyte apheresis achieved a cell yield of 1 × 10^9^ cells for NK cells [62].

Interleukin-2, interleukin-15, and others: In a pilot study, combining interleukin-2 (IL-2) with granulocyte-colony stimulating-factor (G-CSF) showed higher NK cell recovery 14 days post-APBHSCT [63]. In the allogeneic stem cell transplantation, the use of the interleukin-15 (IL-15) super-agonist ALT-803 has been shown to stimulate activation and proliferation of NK cells [64]. Other interleukins that have shown the activation and proliferation of NK cells in vitro, which mean that interleukin-12, interleukin-18, and interleukin-21 could be considered for clinical trials [65].

Plerixafor: Plerixafor is a CXCR4 antagonist approved in combination with G-CSF for the mobilization of stem cells in patients undergoing APBHSCT. CXCL12/CXC4 is a chemokine signal for NK cell trafficking. Multiple studies have shown that the combination of Plerixafor plus G-CSF mobilized more NK cells compared with just G-CSF for mobilization [37,65,66,67,68].

### 9.2. Methods to Sustain NK Cell Recovery Post-APBHSCT

#### 9.2.1. Cytokine-Based Ex Vivo Activation and Expansion of NK Cells

Tarannum et al. provides an excellent review of the different methods of NK cell expansion for adoptive NK immunotherapy [69]. Specifically, in APBHSCT, Nahi et al. reported the feasibility of infusing post-APBHSCT in multiple myeloma patients multiple doses of ex vivo activated and expanded autologous NK cells. The infused NK cells were still detected circulating up to four weeks after the last infusion. The elevation of granzyme B was detected in the bone marrow also 4 weeks after the last NK cell infusion. All patients in the study had detectable responses to the NK cell infusion by measuring the reduction in M-components and/or minimal residual disease [70].

#### 9.2.2. Interleukin-15

Several interleukins have been shown to activate and proliferate NK cells in vitro. A super-agonist complex (ALT-803) of an intelerkin-15 (IL-15) mutein bound to the sushi domain of the IL-15 receptor alpha (IL-15Rα) and fused to the immunoglobulin G1 Fc region has resulted in marked expansion and activation on NK cells in a phase I study as well as in the allogeneic stem cell transplantation setting [64,69]. This provides a platform for the use of ALT-803 in the APBHSCT setting, as our group reported that high IL-15 expression by day 15 post-APBHSCT was associated with higher NK cell recovery and better survival [71].

Figure 1 summaries options during APBHSCT to intervene with NK cell therapies.

## 10. Cancer Stem Cells, Natural Killer Cells and Autologous Peripheral Blood Hematopoietic Stem Cell Transplantation

Cancer stem cells (CSCs) account for tumor resistance to treatment, tumor growth and metastasis, self-quiescence, and renewal [72]. CSCs can evade NK cell anti-tumor activity by downregulating NK-activating NK2G2D and NK cell activator receptors NKp44, NKp30, and CD16 and by increasing KIR receptor ligands [72]. Recent studies have shown the development of the in vitro NK cell expansion of super-charged NK cells (higher cytotoxic cells with high secretion of cytokines) by using sonicated probiotic bacteria AJ2 and osteoclast that can target and eliminate CSCs [73]. This methodology to develop super-charged NK cells is feasible in the APBHSCT by expanding NK cells collected from the autograft and re-infusing them post-APBHSCT to eradicate residual CSCs that survived the high-dose chemotherapy conditioning regimens.

## 11. Conclusions

In this review, I summarized our current understanding of the role of the NK cells that are mobilized, collected, and infused from the autograft and their recovery in patients undergoing APBHSCT. The published reports showed that the infusion of autograft NK as well as early and sustained NK cell recovery directly impact the clinical outcomes post-APBHSCT. The clinical outcome benefits observed based on the NK cells suggest NK cells to be an ideal immune effector target during mobilization, collection, infusion, and recovery/engraftment of NK cells to improve survival in patients treated with APBHSCT.

## Figures and Tables

**Figure 1 cells-11-03469-f001:**
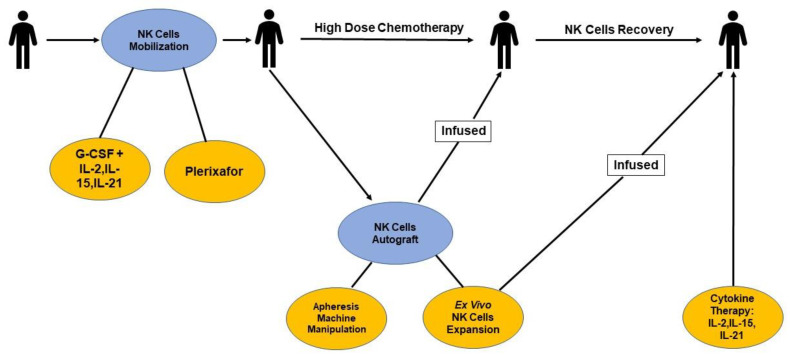
Schematic representation of natural killer (NK) cell graft engineering and recovery in autologous peripheral blood hematopoietic stem cell transplantation (APBHSCT). NK cell mobilization strategies include the use of Plerixafor and cytokines (i.e., IL-2, IL-15, and IL-21) to enhance the mobilization of NK cells in the peripheral blood for harvesting. NK cell autograft strategies include the manipulation of the apheresis machine to harvest high numbers of lymphocytes/NK cells and to harvest NK cells from the autograft for ex vivo expansion. NK cell engraftment/recovery post-APBHSCT includes ex vivo NK cell expansion infusions and the use of cytokines (i.e., IL-2, IL-15, and IL-21).

**Table 1 cells-11-03469-t001:** Summary of clinical outcomes post-autologous peripheral blood hematopoietic stem cell transplantation based on the quantitative and qualitative immune mechanisms of action of natural killer cell.

Disease [Study Reference]	Infused Autograft NK Cell Dose	5-Year Overall Survival Rates	5-Year Progression-Free Survival Rates	*p*-Values
NHL [36]	A-NK ≥ 0.09 × 10^9^ cells/kg	87%		<0.0002
	A-NK < 0.09 × 10^9^ cells/kg	55%		
	A-NK ≥ 0.09 × 10^9^ cells/kg		71%	<0.0001
	A-NK < 0.09 × 10^9^ cells/kg		32%	
NHL [38]	A-NK ≥ 0.5 × 10^6^ cells/kg	70%		0.09
	A-NK < 0.5 × 10^6^ cells/kg	80%		
	A-NK ≥ 0.5 × 10^6^ cells/kg		55%	<0.048
	A-NK < 0.5 × 10^6^ cells/kg		75%	
MM [39]	A-NK ≥ 2.5 × 10^6^ cells/kg	95%		<0.006
	A-NK < 2.5 × 10^6^ cells/kg	65%		
**Disease** **[Study Reference]**	**Infused autograft NK cell receptor dose**	**5-year overall survival rates**	**5-year progression-free survival rates**	***p*-values**
NHL [40]	A-KIR2DL2 < 0.066 × 10^9^ cells/kg	44%		<0.0001
	A-KIR2DL2 ≥ 0.066 × 10^9^ cells/kg	88%		
	A-NKp30 ≥ 0.09 × 10^9^ cells/kg	95%		<0.0001
	A-NKp30 < 0.09 × 10^9^ cells/kg	34%		
	A-NKp30 ≥ 0.09 × 10^9^ cells/kg		77%	<0.0001
	A-NKp30 ≥ 0.09 × 10^9^ cells/kg		13%	
**Disease** **[Study Reference]**	**NK cell absolute number engraftment/recovery days post-APBHSCT**	**5-year overall survival rates**	**5-year progression-free survival rates**	***p*-values**
NHL [42]	Days = 15			
	NK cells ≥ 80 cells/µL	76%		<0.0001
	NK cells < 80 cells/µL	36%		
	NK cells ≥ 80 cells/µL		57%	<0.0001
	NK cells < 80 cells/µL		9%	
MM (non-high-risk cytogenetics) [43]	Days = 30			
	NK cells = 100–200 cells/µL	90%		NP
	NK cells < 100 cells/µL	50%		
NHL [47]	Days = 360			
	NK cells ≥ 250 cells/µL	98%		<0.0001
	NK cells < 250 cells/µL	30%		
	NK cells ≥ 250 cells/µL		95%	<0.0001
	NK cells ≥ 250 cells/µL		10%	
**Disease** **[Study Reference]**	**NK cell receptor absolute numbers engraftment/recovery days post-APBHSCT**	**Cumulative incidence of relapse at 2 years**		***p*-values**
MM (CMV seropositive) [50]	Days = 28			<0.0062
	NKG2C > 1.58 cells/µL	22%		
	NKG2C ≤ 1.58 cells/µL	75%		
**Disease** **[Study Reference]**	**NK cell immunometabolism**	**Progression-free survival 400 days post-APBHSCT**		***p*-values**
NHL/MM [59]	HK2 > 1414 MdFi	25%		<0.001
	HK2 ≤ 1414 MdFi	100%		
	CPT1α > 2830 MdFi	15%		<0.001
	CPT1α ≤ 2830 MdFi	80%		
**Disease** **[Study Reference]**	**KIR and HLA mismatch**		**Clinical outcome prognostic factor**	
AML [53]	KIR and HLA genotypes of low-affinity interactions		Relapse	
NHL [56]	HLA-Cw8 genotype		Overall survival	
Neuroblastoma [55]	KIR-HLA mismatch		Disease progression	
MM [54]	KIR3DS1 genotypes		Progression-free survival	

Abbreviations: AML = acute myelogenous leukemia: A-NK = Autograft natural killer cell; HK2 = hexokinase 2; CMV = cytomegalovirus: CPT1α = carnitine palmitoyl-transferase I alpha; HLA = human leukocyte antigen; KIR = killer-cell immunoglobulin-like receptor; MdFi = modified fluorescence intensity; MM = multiple myeloma; NHL = non-Hodgkin’s lymphoma; and NK cell = natural killer cell.

## Data Availability

Not applicable.

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
