# Peer review of "Natural Killer Cells Are Key Host Immune Effector Cells Affecting Survival in Autologous Peripheral Blood Hematopoietic Stem Cell Transplantation"

_cells, 2022, doi:10.3390/cells11213469_

Round 1
Reviewer 1 Report
Overal summary: The author presents a very complete review on the role of NK cells in autologous stem cell transplantation (ASCT). The subject fits the journal scope and has a great potential for readers’ interest. The manuscript is written appropriately, and minor English language mistakes were found. Although based on old concepts of immunosurveillance, the general concept of the manuscript is novel in the sense of highlighting the potential of ASCT as immunotherapy. The author reviews the overall benefit of the graft content and recovery of NK cells in ASCT, and encourages strategies to enhance collection and sustain NK effects in this setting. There is no doubt that the author is an expert in this area of research but, the weakness of the manuscript is redundancy with previous works and publications that resulted in 20 self-citations.
General comments: As mentioned, the review is very complete and updated on the covered topic. The conclusions drawn are coherent and supported by the listed citations. A major concern regarding self-citation needs to be pointed out to follow with the Journal’s guidelines. The author has very recently published a review in this Journal addressing the immune cell autograft content of dendritic cells, T cells and NK cells globally (Porrata LF. The Impact of Infused Autograft Absolute Numbers of Immune Effector Cells on Survival Post-Autologous Stem Cell Transplantation. Cells. 2022 Jul 14;11(14):2197). The concept of this new submission is the same, i.e, highlights the immunotherapeutic potential of autologous stem cell transplantation besides its purpose to reestablishing hematopoiesis after high dose chemotherapy, but focuses on NK cells. Indeed, this is at least the fifth review of data or literature from the author on the same subject.
Items 4, 5, 6 of the text consist most entirely of the author’s own work. These data could be summarized and directly address NK cells data only, preferentially avoiding the frequent self-citation. Items 7, 8, 9, 10 present reviews from other perspectives and a proposal for NK cells graft engineering and recovery that shows a different and novel and different contribution. Although the authors previous contributions are relevant, the data presented is redundant, already available for the scientific community.
Specific comments:
Figure 1 summarizes the NK cells’ mechanisms of action to improve autologous stem cell transplantation outcomes. As it is, the up and down arrows of the figure are much greater than the mechanisms themselves. A table would show the data more properly or the figure could be modified. Also, the fourth mechanism (NK cells engraftment/recovery) is not mentioned in the figure legend.
Some phrases need improvement in order to clarify the statements. A review is recommended. For example:
…the immunomodulatory of the NK cells – line 93
…enhancing the dendritic cells to maturation and presentation of the antigens – line 99
…NK cell activity can returned – line 144
Day 14 after autologous stem cell transplantation 145 were collected – line 145
Author Response
Please PDF file attached

Reviewer 2 Report
This review focuses on updating the field on the role of the NK cells in autologous peripheral blood hematopoietic stem cell transplantation (APBHSCT) by an expert in the area. The author includes, summarizes, and discusses the most current data regarding the association of autologous NK cells and clinical outcomes in patients post-APBHSCT.
The timing of this review is ideal, given the recent developments in the field over the past five years. Notably, the clinical scope of this review is significant because it does not seem to overlap with a recently published review by the same author (10.3390/cells11142197) or by other groups (10.3390/cancers13071589). The manuscript is generally well-written and follows a logical structure. Notably, it includes many clinical trial references relevant to the main topic.
Because this review aims to update the reader on this topic, the manuscript should consider using more up-to-date references in Section 2: Natural killer cell biology and the section on B- and T-cell reconstitution after APBHSCT. Finally, the manuscript would benefit from avoiding statements of novelty or priority wherever possible.
Although Figure 1 summarizes the main NK-cell mechanisms of action associated with positive and negative outcomes post-APBHSCT, the manuscript would benefit from including a Table summarizing the relevant results from the clinical trials cited throughout the text.
Minor comments:
Line 50: there is a typo; it should say NKG2A instead of NKG2C.
Line 65: The definition for an "adaptive NK cell" is missing.
Lines 128 and 132: the statements lack references.
Line 283: missing word.
Line 319: CD56bright/dim NK cell nomenclature should be used consistently (see line 37).
Line 325: the author's name should be Arteche-Lopez instead of Lopez.
Line 424: the author should avoid referencing unpublished data that has not been peer-reviewed.
Line 438: this statement lacks a reference, while reference #75 is not included in the main text.
Line 439-440: plerixafor is a CXCR4 antagonist.
Line 454: the interleukin number is missing; a new paragraph should come after this line.
The reference list contains typos (Line 515, 539, 564).
Figure 1: Spelling of 'NK cells.'
Figure 2: The right side looks cut off in the PDF.
Author Response
Pease see PDF file attached

Round 2
Reviewer 1 Report
The manuscript has improved substantially with the review. The author has followed the suggestions regarding one of the figures and rewrote some sentences to clarify the text. Also, he minimized the self-citations by removing one of the topics and justifies the use of other self-citations: "I took the liberty to use few other self-citations for these items to keep the flow and logical structure of the manuscript." Indeed, the text is well written and scientifically sound, although the self-citations are still present (they were reduced from 20 to 17).
Author Response
I would like to thanks the reviewer for making the manuscript stronger and better and for the reviewers time.